# Numerical Reconstruction Model and Simulation Study of Concrete Based on Damaged Partition Theory and CT Number

**DOI:** 10.3390/ma12244070

**Published:** 2019-12-06

**Authors:** Jianyin Fang, You Pan, Faning Dang, Xiyuan Zhang, Jie Ren, Na Li

**Affiliations:** School of Civil Engineering and Architecture, Xi’an University of Technology, Xi’an 710048, China

**Keywords:** concrete, damaged partition, CT test, numerical simulation, model reconstruction

## Abstract

The applicability of mesoscopic models plays an important role in studying the mesoscopic mechanical properties of concrete. In this study, the computerized tomography (CT) test of concrete under uniaxial compression conditions is conducted using a portable dynamic loading equipment developed by Xi’an University of Technology and a medical Marconi M8000 spiral CT scanner. On the basis of damage partition theory, a probabilistic statistical method for determining threshold values is proposed, and a CT test images is obtained and divided into aggregate, hardened cement and hole-crack areas. A ‘structural random numerical concrete model’ is also established on the basis of the coordinates of each pixel unit in CT images. Uniaxial static compression and tensile numerical simulation tests are conducted. Results show that the structural random numerical concrete model can not only reflect the microscopic composition of concrete but also the interfacial transition zone (ITZ) between aggregate and mortar. The ITZ thickness is approximately 0.04 mm, which is close to the real concrete sample ITZ thickness (approximately 10–50 μm). In the two tests, the specimen damage starts from the initial defects, and the damage crack expands through the weaker ITZ around the aggregate. No matter under the action of static tension or compression load, the damage cracks of the sample almost never pass through the aggregate. Most of the many cracks in uniaxial compression are shear cracks. However, many cracks form at the beginning of uniaxial tension, and only one main crack, which is roughly perpendicular to the loading direction, exists in the end.

## 1. Introduction

Since the proposal of ‘numerical concrete’ by Roelfstra et al., meso-scale numerical models based on finite element (FE) analysis have been widely used in the research of rock and concrete [1]. However, in these models and methods, concrete is often simulated as an aggregation of aggregate (coarse and fine aggregates), cement mortar, and the interfacial transition zone (ITZ) between them at the mesoscopic level [2,3]. Admittedly, the shape, size and spatial distribution, and volume of aggregates within the mortar matrix significantly influence the mechanical behaviour of concrete. Therefore, it is vital to represent aggregates in numerical models as realistic as possible when the behaviour and fracture process of concrete are modelled since the aggregates may dominate the damage accumulation and crack propagation at the mesoscopic level.

To date, a number of efforts have been devoted to generating realistic aggregate shape, size distribution and spatial distribution in numerical simulations of concrete behaviour. Many methods have been developed based on advanced computer technology to realistically simulate aggregates in 2D and 3D. In 2D numerical models, aggregates are often simulated as having circular shapes [4,5,6,7], angular or polygonal shape [8,9,10] or even arbitrary shapes [11]. In 3D numerical models, however, aggregates are frequently assumed to be spherical shape [12,13,14,15,16] or ellipsoidal shapes [17,18].

Although the virtually-generated numerical concrete by the methods mentioned above may have the same internal structure as real concrete in terms of size distribution and volume of aggregate, its limitations are also obvious in terms of the spatial distribution and shapes of aggregates. Therefore, it is necessary to develop innovative methods to more accurately model the micro/meso structures of concrete.

However, CT scanning techniques can obtain the 3D internal structure of objects using X-rays. Thus, these techniques have been routinely applied in hospitals. CT scanning technology has become a powerful tool for describing the mesoscopic structure of materials due to its high-resolution, non-destructive and clear 3D visualisation properties. In the past decade, researchers have successfully applied this technology to characterise mesoscopic structure and study damage evolution of various materials, such as Man and Mier [19,20,21] mapped the realistic aggregate information of concrete microstructure obtained by CT technology into the lattice model to study the size effect of concrete by focusing on its strength and fracture energy. Ren et al. [22] converted the XCT-images of concrete to finite element (FE) meshes using the commercial packages AVIZO, and achieved limited success in their 2D cohesive fracture modelling. Huang et al. [23] developed a novel 3D XCT-image based meso-scale FE fracture modelling method to simulate complicated damage initiation and evolution in concrete using the concrete damage plasticity (CDP) model of ABAQUS. Mostafavi et al. studied the damage of polycrystalline graphite [24] region. Sharma studied the ITZ damage of carbon/carbon composite [25]. Jivkov et al. simulated the brittle damage of concrete.

In fact, according to the CT imaging principles [26], various artefacts exist in CT images due to imaging physics and geomaterial complexity. However, there are few existing commercial packages able to process CT images, especially those of geomaterials, which limits the application of the CT technique in characterizing heterogeneous geomaterials such as concrete. In addition, as the transition zone between aggregate and cement mortar, the strength of interface determines the mechanical properties of concrete and the mode of crack propagation [27,28,29].

To this end, a CT test of concrete under uniaxial compression conditions is conducted. In accordance with the test results, a concrete scanning pattern is quantitatively partitioned on the basis of damage partition theory. The method of determining partition threshold is studied. Then, ‘structural random numerical concrete model’, a 2D mesoscale FE model, is developed. By using the secondary development of ANSYS software, the numerical simulation of concrete under uniaxial compression and uniaxial tensile stress is performed by introducing the double-fold damage constitutive model. Finally, the crack evolution of concrete under tensile and compressive stress is studied. 

## 2. Uniaxial Compression CT Test

### 2.1. Portable Power Loading Equipment

This test adopts the newly developed portable power loading equipment of Xi’an University of Technology, which is the first power loading test equipment with CT scanners in China [30]. The CT scanner uses a Marconi M8000 spiral CT scanner with an image size of 1024 × 1024. The maximum imaging speed is 0.5 s for four-layer scanning. The loading scanning device is shown in Figure 1.

### 2.2. Test Conditions and Processes

#### 2.2.1. Test Conditions

This test uses a first-grade C15 concrete cylinder test piece with a size of Φ60 mm × 120 mm, a water-cement ratio of 0.4 and an aggregate particle size of 5–20 mm. Samples are cured for 28 days under standard conditions, and the test is then conducted.

#### 2.2.2. Test Processes

During the test, the concrete specimen is placed between the two pistons in the loading device, and a small force is initially applied. In this manner, the specimen does not slip. Then, the portable loading device is placed in the centre of the CT scanner through the hospital bed, and the initial scanning is performed. The initial scan results are used as a comparison between the loading stages of the scan. The loading is stopped every scan, and the loading is repeated in accordance with the loading, scanning, reloading and scanning sequences until the concrete breaks. Displacement loading method is adopted, and the loading rate is 0.002 mm/s. The scanning is performed 11 times. The number of scanning layers of the sample is 70 longitudinal layers and 24 transverse layers. Each scanning layer is evenly arranged in accordance with the sample. The scanning position and the stress–strain curve are shown in Figure 2. Two representative sections are selected for the study. Seven representative scans are selected for each section, as shown in Figure 3.

The larger the CT number in the CT image, the higher the brightness, and the greater the density of the representative material. The brighter areas in the Figure 3 are aggregates (coarse and fine aggregates), the grey areas are mortars (products of cement hydration hardening) and the black areas are holes and cracks.

## 3. Damage Evolution Theory 

### 3.1. Integrity and Damage Degree

On the basis of fuzzy mathematics theory, the entire CT scan image is called the whole field, which is represented by a set as Ω = {(*x*, *y*, *z*)|(*x*, *y*, *z*) is an arbitrary point on the study object space area}. Any point in the whole field is complete, but the integrity is different. On this basis, the integrity can be defined as follows [31]:(1)P(x,y,z)=[H(x,y,z)+1000]/[maxH(x,y,z)+1000]
where the *H*(*x*, *y*, *z*) is the CT number of the space point (*x*, *y*, *z*), which can be defined as follows:(2)$$H\left(x,y,z\right)=\frac{\mu_{t}-\mu_{w}}{\mu_{w}}\times 1000$$
where $\mu_{t}$ and $\mu_{w}$ are the X-ray linear attenuation coefficients of minerals and water in the scanned image, respectively.

As opposed to integrity, the damage degree is defined as:(3)d(x, y,z)=1−P(x,y,z)

According to this definition, integrity and damage degree are in the interval of [0, 1]. Figure 4 shows the image of the integrity and damage degree of the same CT scan of concrete.

### 3.2. λ Level Complete Domain and λ Level Damage Domain of Sample

Assuming that 0≤λ≤1, set:{(x,y,z)|(x,y,z)∈Ω,andλ≤P(x,y,z)≤1}
is defined as the *λ* level complete domain of sample, expressed by *P_λ_.*

Moreover, set:{(x,y,z)|(x,y,z)∈Ω,andλ≤d(x,y,z)≤1}
is defined as the *λ* level damage domain of sample, expressed by *d_λ_*.

Figure 5 shows the complete domain and damage domain of the concrete at different *λ* levels. The *λ* level damage domain represents holes and cracks. Different *λ* values represent cracks of different scales. Therefore, the selection of *λ* can achieve the unity of macro and micro.

### 3.3. $\left(\lambda_{1}-\lambda_{2}\right)$ Intercepted Sections

To achieve partitioning of concrete aggregate, mortar, ITZ and hole cracks, $\left(\lambda_{1}-\lambda_{2}\right)$ intercepted sections can be defined to describe it.

Assuming that $0\leq \lambda_{1}\leq \lambda_{2}\leq 1$, set:$$\left\{\left(x,y,z\right)|\lambda_{1}\leq d\left(x,y,z\right)\leq \lambda_{2},\left(x,y,z\right)\in \Omega ,0\leq \lambda_{1}\leq \lambda_{2}\leq 1\right\}$$
or:$$\left\{\left(x,y,z\right)|\lambda_{1}\leq P\left(x,y,z\right)\leq \lambda_{2},\left(x,y,z\right)\in \Omega ,0\leq \lambda_{1}\leq \lambda_{2}\leq 1\right\}$$
is defined as $\left(\lambda_{1}-\lambda_{2}\right)$ complete domain intercepted sections or $\left(\lambda_{1}-\lambda_{2}\right)$ damage domain intercepted sections of the sample section, expressed by $d_{\lambda 1-\lambda 2}$ or $P_{\lambda 1-\lambda 2}$. Figure 6 shows the partition intercepted sections of a scan section of a concrete sample. The intercepted sections can distinguish well the materials of different densities in the concrete, such as aggregates, cement mortar and cracks.

The descriptions of integrity, damage degree, complete domain, damage domain and intercepted sections are relatively simple, and further detailed descriptions can be found in the literature published by the author in the Journal of Hydroelectric Power [32].

## 4. Study on Quantitative Partition of Concrete CT Test Process

On the basis of the definition of intercepted sections, the concrete scanning section is divided into aggregate, hardened cement and hole–crack areas.

### 4.1. Quantitative Partition of Concrete CT Image

On the basis of the definitions of complete domain and $\left(\lambda_{1}-\lambda_{2}\right)$ intercepted sections, specific partitions are defined as follows.

When $0\leq P\left(x,y,z\right)<\lambda_{1}$, the density of the material is small. On this basis, the material contains holes or breaks, which are called hole–crack area, which is recorded as $P_{0-\lambda 1}$.

When $\lambda_{1}\leq P\left(x,y,z\right)<\lambda_{2}$, the density of material is not excessively large, and the integrity of material is general. On this basis, the material is in the hardened cement area, which is recorded as $P_{\lambda 1-\lambda 2}$.

When $\lambda_{2}\leq P\left(x,y,z\right)<1$*,* the density of material is relatively high, and the integrity is high. On this basis, it is in the aggregate (coarse and fine aggregates) area, which is recorded as $P_{\lambda 2-1}$.

The division is based on the CT number. Thus, the fine aggregate in the mortar is also divided into the aggregate area in the division, whereas only cement hydration and hardening products remain in the cement mortar. In this study, these products are called ‘hardening cement’.

### 4.2. Using Probability Statistics Method for Determining Partition Thresholds λ_1_ and λ_2_

The focus of the partition is how to determine reasonable partition thresholds *λ*_1_ and *λ*_2_. For homogeneous rocks, *λ*_1_ and *λ*_2_ can be obtained using maximum-minimum method. However, for concrete, this method cannot be used. This method is proposed based on homogeneous materials. Hence, its applicability to heterogeneous materials, such as concrete, is limited. Therefore, the method of probability statistics is used to determine the threshold.

Concrete is a composite material composed of aggregates, mortar and ITZ. The idea of partition indicates that a component of its resolution unit should be continuous and gradually distributed to avoid large jump changes. The distribution of the resolution units among different components is bound to a certain jump. Therefore, a total of 137,027 resolution units are required to select a CT scan as a universe Ω. The statistical relationship between the threshold and the resolution unit is shown in Figure 7, where the abscissa is the threshold and the ordinate is the corresponding number of statistical units.

As shown in Figure 7, when the thresholds are approximately 0.7 and 0.84, the number of statistical resolution units has a large jump; therefore, *λ*_1_ and *λ*_2_ are preliminarily 0.7 and 0.84, respectively.

To improve the accuracy of the threshold selection, two sub-areas are selected for the statistical analysis of *λ*_1_ and *λ*_2_, whose threshold centres are on 0.7 and 0.84, respectively. As shown in Table 1, the thresholds for the large jump of statistical resolution units are 0.698 and 0.843. Therefore, thresholds *λ*_1_ and *λ*_2_ are set as 0.698 and 0.843, respectively.

## 5. Establishment of a Structural Random Numerical Concrete Model

### 5.1. CT Image Acquisition and Image Partition Processing

The reconstruction of the structural random numerical concrete model is based on the initial CT scan image of the uniaxial compression CT test, as shown in Figure 8.

The first task of modelling is to perform image localisation. Then, the CT number of CT images is extracted. The coordinates of the resolution unit in the CT scan image were used to determine the location of the modeling image, and the selection point of the graph coordinates was judged according to the CT value of the resolution point, The positioning accuracy of the image is one CT pixel (approximately 0.009 mm). Following the concept of integrity (*P*(*x*, *y*, *z*)), the extracted CT numbers are normalised. Then, following the concept of intercepted sections, the CT scan image is divided into aggregate, hardened cement and hole–crack areas, which are marked as *P*_0–λ1_, *P*_λ1–λ2_ and *P*_λ2–1_, respectively.

The images of each subarea of the concrete CT image obtained by the above method are shown in Figure 9.

This method divides the gravel with large density and size in the sand into the aggregate. In this manner, the numerical reconstruction model can effectively reflect the actual situation of the material, and the simulation effect can be ideal.

The reconstructed model is shown in Figure 10.

In the whole modelling process, the data calculation is realised by using Fortran program compiled by ourselves.

### 5.2. Numerical Experiment Based on the Structural Random Numerical Concrete Model

#### 5.2.1. Conditions of the Numerical Test

According to Lemaitre equivalent strain principle, the stress-strain relationship can be expressed as:*σ* = *Eε*(4)
where *σ* is the stress, *E* is the modulus of elasticity after damage, *ε* is the strain.
*E* = *E*_0_(1 − *D*)(5)
where *E*_0_ is the initial modulus of elasticity, *D* is the damage variable [33]:(6)$$D=\{\begin{array}{c} 0\;\varepsilon_{max}<\varepsilon_{0}\;\\ 1-\frac{\eta -\lambda }{\eta -1}\frac{\varepsilon_{0}}{\varepsilon_{max}}+\frac{1-\lambda }{\eta -1}\;\varepsilon_{0}<\varepsilon_{max}\leq \varepsilon_{r}\;\\ 1-\lambda \frac{\varepsilon_{0}}{\varepsilon_{max}}\;\varepsilon_{r}<\varepsilon_{max}\leq \varepsilon_{u}\;\\ 1\;\varepsilon_{max}>\varepsilon_{u}\; \end{array}$$

λ is strength residual coefficient, *ε*_0_ is the main tensile strain corresponding to the tensile strength, *ε**_r_*is the tensile residual strain:(7)$$\varepsilon_{r}=\eta \varepsilon_{0}$$

η is the residual strain coefficient, *ε**_u_* is the ultimate tensile strain:(8)$$\varepsilon_{u}=\zeta \varepsilon_{0}\left(\zeta >\eta \right)$$

ζ is the limit strain coefficient, $\varepsilon_{max}$ is the maximum value of the main tensile strain in the loading history.

According to literature [33], λ is set as 0.1, η as 5, and ζ as 10.

For brittle materials, such as concrete, both tensile failure and compressive failure are ultimately due to the fact that the tensile stress exceeds its ultimate tensile strength. Therefore, the maximum tensile strain strength criterion is adopted in this paper, that is, the element starts to damage when the maximum principal tensile strain value of the element exceeds the given limit tensile strain threshold, and the element still has a certain bearing capacity. The damage evolution model is shown in Equation (6).

Generally, material parameters determine the accuracy of the numerical test results. Hence, to maintain consistency in physical and numerical test results, reasonable physical test parameters must be used. In accordance with the physical tests, the material parameters as shown in Table 2 can be obtained. The kill unit function in ANSYS is used to realise holes.

When the unit’s maximum tensile strain reaches $\varepsilon_{u}$, the unit is considered completely destroyed and then killed (through the unit failure function to achieve).

To match the numerical simulation test results with the physical CT test, the constraint conditions in the numerical simulation are the same as those in the physical test. As such, the bottom surface is fully constrained and the top surface is horizontally constrained. Displacement loading is used for loading.

Displacement loading method is adopted in this test, and the loading steps were consistent with CT test. Each loading is 2 × 10^−4^ m, and the loading stops when the sample is broken.

#### 5.2.2. Static Uniaxial Compression Numerical Test

The results of the tests based on the above conditions are shown in Figure 11 and Figure 12.

As shown in Figure 11, the change of the axial displacement under the uniaxial compressive load is relatively uniform at first. As a result, the displacement at both ends is larger, whereas that at the middle is smaller. With the increase in the load, there is a strong non-uniformity of displacement. With the further increase in displacement loading, the specimen shows damage, and the internal of the specimen begins to show fracture cracks. At the same time, the internal stress of the specimen is redistributed. Meanwhile, the axial displacement in the specimen appears localised.

The change law of horizontal displacement (Figure 11) is similar to the physical test phenomenon. At the beginning, with the increase in the axial displacement loading, the specimen deformed laterally, and the horizontal displacement of the specimen developed symmetrically to both sides. However, because of the inhomogeneity of the sample and the presence of initial defects, the horizontal displacement develops asymmetrically with the increase in the load, and the asymmetry of the displacement becomes evident until the sample is damaged. Later, due to the generation of damage cracks, the stress in the sample redistributes and the stress concentration becomes evident. At this time, the horizontal displacement of the sample shows a strong locality (Figure 11; fifth to seventh loads in Figure 12).

The main stress nephogram of each loading stage of sample under uniaxial compression loading is shown in Figure 12. Figure 12 can directly reflect the stress localisation of the sample under the action of pressure load. At the beginning of loading, the initial defects in the sample and the contact part between the aggregate and the cement mortar are prone to stress concentration, and nearly all parts of the sample show evident stress concentration. However, as the load increases, stress concentration areas no longer widely existing, and stress is concentrated in areas where local defects are serious. This phenomenon occurs because, as load increases, the stress concentration area in the sample begins to produce damage and redistribute stress, which results in evident local stress concentration phenomenon, that is, these areas will initially produce damage cracks. This phenomenon can be observed in the stress nephograms of the fourth loading. With the stress concentration, damage cracks begin to appear in these areas. The occurrence of damage cracks makes the strain energy of the sample released to a certain extent. Meanwhile, due to the existence of cracks, the crack tip effect becomes evident, thereby intensifying the local concentration of stress (sixth and seventh loading in Figure 12). Then, the damage crack occurs, and the specimen loses its bearing capacity.

The law of crack propagation has always been a topic of interest in the research on concrete, and its analysis can reveal the intrinsic mechanical properties of concrete from the root. Figure 13 shows a crack growth diagram of the specimen under uniaxial compression loading.

As shown in Figure 13, under uniaxial static compression loading, the stress concentration easily occurs in the aggregate tip and initial defects. Therefore, the damage and failure of concrete specimens usually start from initial defects and produce initial cracks at multiple points. The red area indicates a crack. The damage crack in the third loading is relatively small. However, with the increase in load, the crack expands and passes through and further forms a macroscopic crack. The strength of the ITZ and the hardened cement is relatively weak. As a result, the crack generally grows around the aggregate at the ITZ between the aggregate and the hardened cement (the loading crack diagram of the fourth loading in Figure 13). From the mechanism analysis, cracks hardly form through the aggregate in the static load, primarily because the aggregate has higher strength than the cement mortar. Hence, the cracks have sufficient time to expand along the weak plane (the bonding surface of aggregate and cement mortar) in the sample under static load. The specimen is not destroyed immediately after the main crack is formed under static pressure. Stress recombination occurs in the sample, which results in further cracks in other areas of the specimen. Therefore, the specimen still has a certain residual strength after the main crack is generated, which shows a strong softening property. Under compressive load, the specimen cracks are mostly shear cracks, which are approximately 45° to the axial direction of the specimens, and the number of cracks is numerous; this result is consistent with the results of CT test of uniaxial static compression of concrete (Figure 3).

In accordance with the above analysis, simulating the damage evolution process of concrete under uniaxial compression by using the structural random numerical concrete model and the damage evolution equation of double broken line is suitable. This method can also effectively reflect the damage and failure mechanism of the interior of concrete under compression.

#### 5.2.3. Static Uniaxial Tension Numerical Test

In the uniaxial tensile test, except for the loading direction, the other conditions and parameters are the same as that of uniaxial compression test. A total of 10 loading tests are performed with the displacement as the loading method. The test results are shown in Figure 14, Figure 15 and Figure 16.

Figure 14 shows the longitudinal displacement nephogram under uniaxial tensile loading. During the first to third loading process, the displacement is uniform. The displacement is large at the two ends and small at the middle of the sample, and the displacement is uneven in the initial defects, such as holes and cracks. During the fourth loading, the longitudinal displacement of the sample begins to show non-uniformity, and the phenomenon of large local displacement occurs at the large initial defects. This result fully reflects the localised characteristics of the deformation of the sample. As the loading displacement at both ends continues to increase, the local deformation phenomenon in the specimen is intensified during the fifth to tenth loading processes, and the displacement is uniform in other places. This result is similar to the displacement variation characteristic under uniaxial compression conditions. The root cause is that a damage crack is generated in the sample during this process, and the strain energy in the sample is released as a result of the crack. Therefore, the displacement is more uniform in other places. The occurrence of cracks further exacerbates the stress concentration at the crack tip, which causes cracks to propagate further through with the action of tip effect and forming a macroscopic crack. The penetrating macroscopic cracks exacerbate the localised effect of the strain. Hence, the local characteristics of the displacement at longitudinal becomes increasingly evident with loading.

In view of the stress, the stress distribution of the specimen under uniaxial tensile load is similar to that under uniaxial compression loading (the first loading in Figure 15), and the stress concentration at the ITZ of aggregate and hardened cement in the specimen and initial defects is common during initial loading. However, with the increase in load, the stress concentration in the middle of the specimen is simpler than that under uniaxial compression because the damaged surface of the specimen in the uniaxial compression test is far more than that in the uniaxial tensile test. With the increase in the loading displacement, the stress concentration phenomenon appears in many places in the specimen (the fourth loading in Figure 15). After the fifth loading, the stress concentration range in the sample has a relatively large reduction compared with the fourth loading because the penetration of the crack inhibits the growth of microcracks in other parts, which results in the intensification of stress concentration at the macrocrack. This result is also a sign of the destruction of the sample, which indicates that major cracks will then occur in these areas. This finding can be confirmed from the crack distribution at the tenth loading in Figure 15.

Figure 16 shows the evolution law of cracks under tensile loads. Before the third loading, the load does not exceed the ultimate bearing capacity of the specimen material. Hence, no cracks form in the specimens during the first and second loading. In the third loading, a small number of microcracks in the sample began to appear at the initial defect. Thus, the crack initiation is controlled by the initial defect, which is consistent with the conclusions obtained by many researchers. As the loading displacement continues to increase, microcracks begin to appear in many places in the specimen (the red area in the Figure 16) Therefore, under uniaxial tensile loading, multiple initial cracks develop simultaneously in the specimen. As shown in Figure 16, many microcracks propagate simultaneously however, at the end of the test only one major crack is present because of the occurrence of major cracks that inhibit the propagation of other secondary cracks. From the perspective of the path of crack propagation, the crack initiates from the initial defect and then propagates along with the relatively weak ITZ around the aggregate. This is because the static loading rate is relatively slow, and the crack has enough time to expand and complete in the weak material. During the loading process, with the generation of the main crack, the strain energy of the material is released, thus inhibiting the generation of other cracks, resulting in the crack growth hardly passing through the aggregate. The main crack is approximately perpendicular to the loading direction of the sample, with a small number of cracks. This result is consistent with the conclusion obtained in the concrete CT test under static uniaxial tensile load.

In accordance with the above analysis, the CT test using the concrete numerical model of stochastic structure can also reflect the stress distribution and crack evolution process of the concrete specimen under uniaxial tension condition.

## 6. Conclusions

This study uses the damaged partition method to reconstruct the structure random numerical concrete model, with the pixel points of the CT scan as the unit. The numerical simulation test is compared with the physical CT test. The following conclusions can be drawn:
(1)The method of determining the threshold using probability statistics method is proposed. The concrete CT scan is divided into aggregate, hardened cement and hole-crack areas (initial defect) in accordance with different thresholds. The integrity of CT image information is ensured, thereby giving full play to the value of CT numbers of each resolution unit.(2)On the basis of the coordinates of each pixel unit in the CT scan, the structural random numerical concrete model is established using the secondary development of ANSYS software. The model is similar to the actual concrete sample. It not only can reflect the mesoscopic composition of concrete but also the bonding ITZ between the aggregate and the hardened cement. The thickness of the ITZ established by the model is approximately 0.04 mm, which is close to the real ITZ thickness (approximately 10–50 μm) and considerably smaller than the ITZ (greater than 1 mm) formed by using the random aggregate model. As such, the numerical tests show that the simulated mechanical properties of concrete are authentic.(3)The numerical simulation tests results show that in all tests, the damage crack of concrete sample experienced the process of initiation, slow expansion, slow penetration and sudden rapid penetration, the damage of the specimen begins from the initial defect and the damage cracks then propagate along the weak ITZ around the aggregate. Most of the many cracks that form in the uniaxial compression test are shear cracks. In the uniaxial tension test, many cracking points form at the beginning and multiple cracks develop simultaneously. However, only one major crack is present at the end of test, which is roughly perpendicular to the loading directions. No matter under the action of static tension or compression load, the damage cracks of the sample almost never pass through the aggregate, which is consistent with the results of physical CT test.(4)This study results can provide a certain methods and theoretical basis to static and dynamic characteristics research of mass concrete. The study only established the mesoscopic structure of concrete from a two-dimensional perspective, cannot study the concrete from the three-dimensional perspective. The next study should be establish three-dimensional concrete numerical model based on two-dimensional concrete structure, and study the effect of material heterogeneity to the static and dynamic characteristics of mass concrete based on CT number.(5)The calculation results of the traditional model have significant mesh size dependence in the numerical simulation study, and the regularization method is often used to correct it. However, the mesh dependence problem is not considered in the numerical simulation, which is a defect of this study, the authors are going to focus on the analysis and solution of this problem in subsequent research.

## Figures and Tables

**Figure 1 materials-12-04070-f001:**
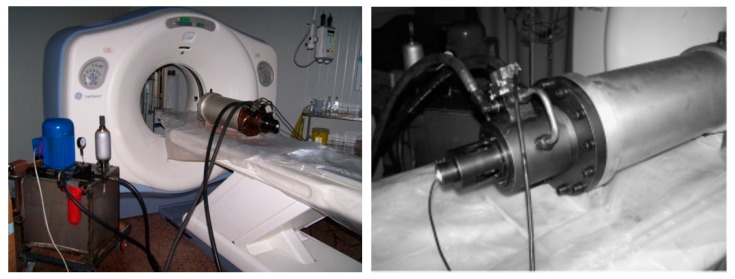
Portable dynamic loading equipment and CT scanner.

**Figure 2 materials-12-04070-f002:**
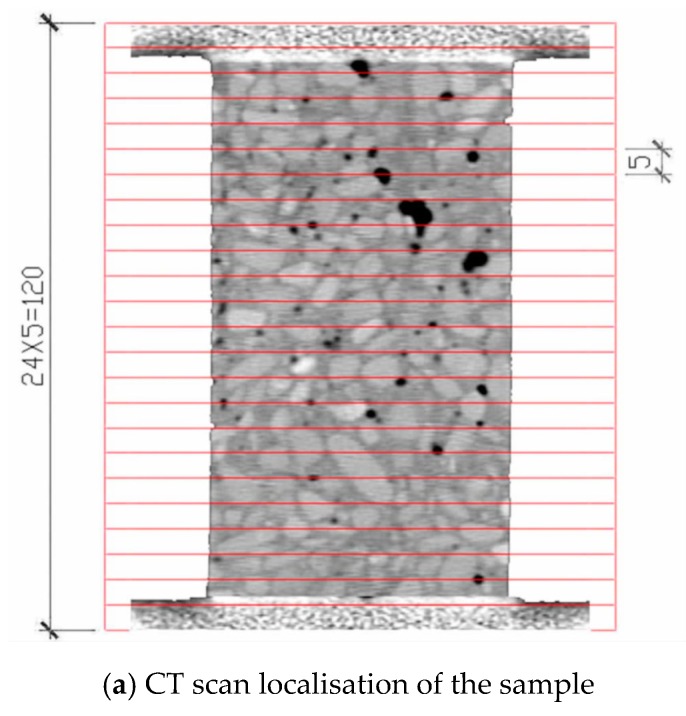
CT scan localisation and load–time curve.

**Figure 3 materials-12-04070-f003:**
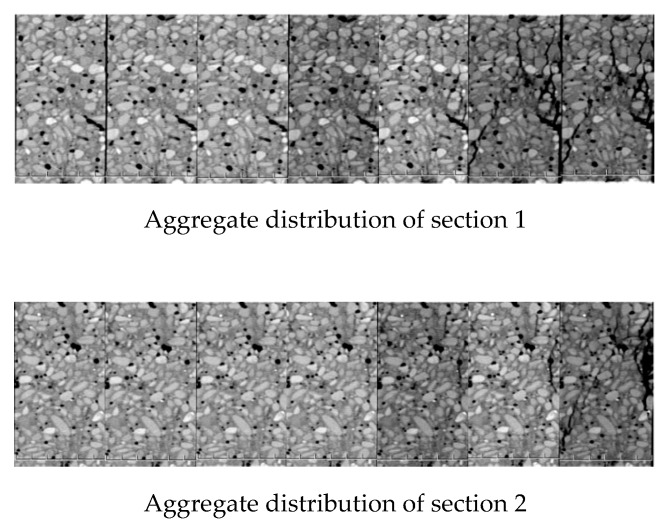
CT scans.

**Figure 4 materials-12-04070-f004:**
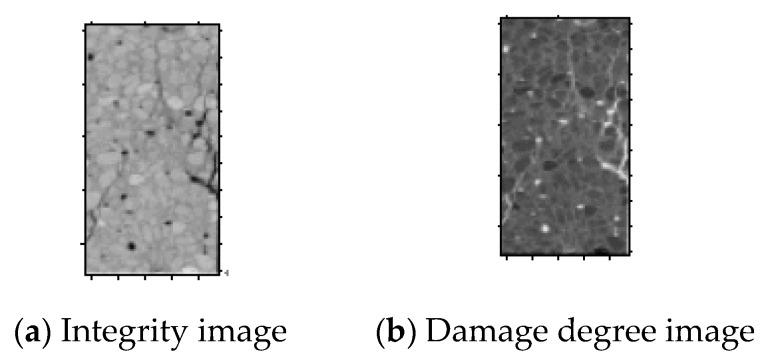
Striograph.

**Figure 5 materials-12-04070-f005:**
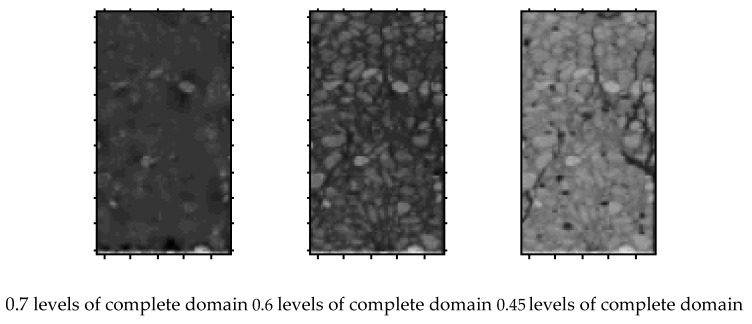
Different λ levels of complete and damage domains.

**Figure 6 materials-12-04070-f006:**
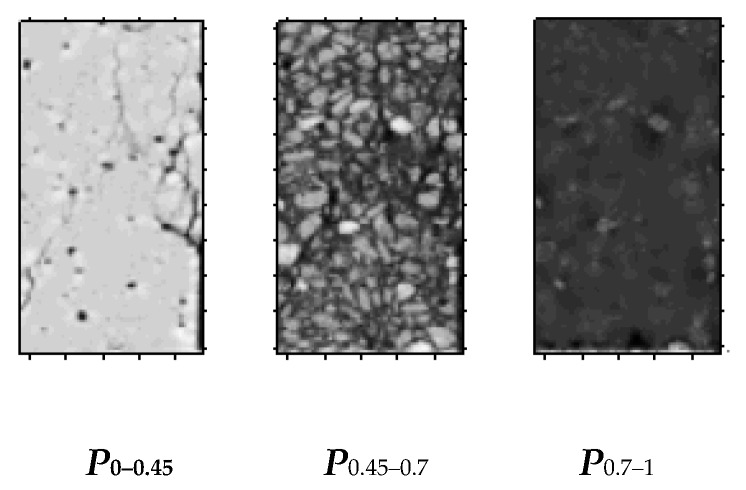
Intercepted sections of perfect fields.

**Figure 7 materials-12-04070-f007:**
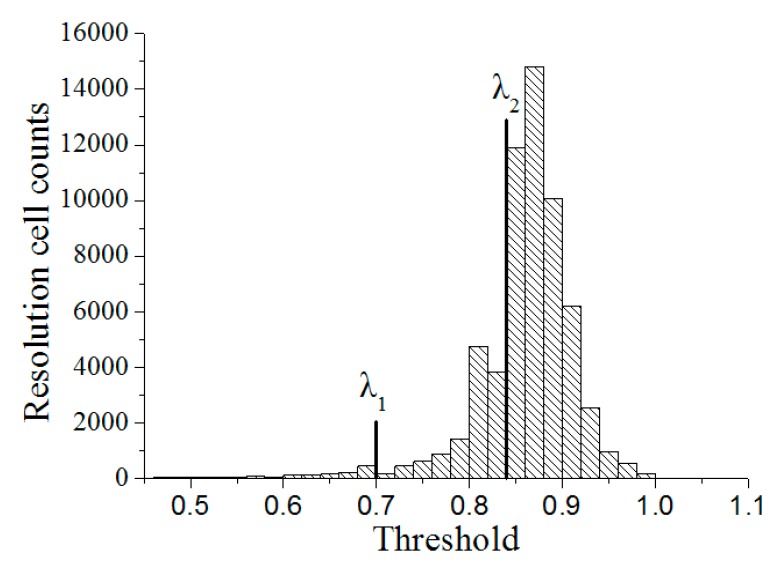
Relationship of threshold and resolution cell.

**Figure 8 materials-12-04070-f008:**
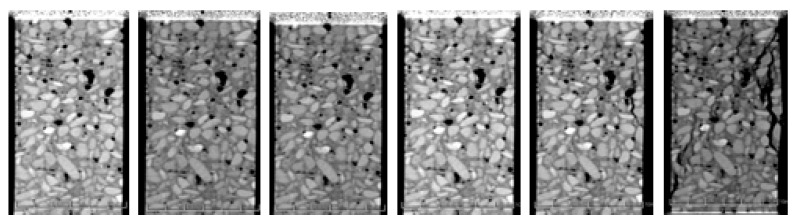
Scans of the uniaxial compression CT test.

**Figure 9 materials-12-04070-f009:**
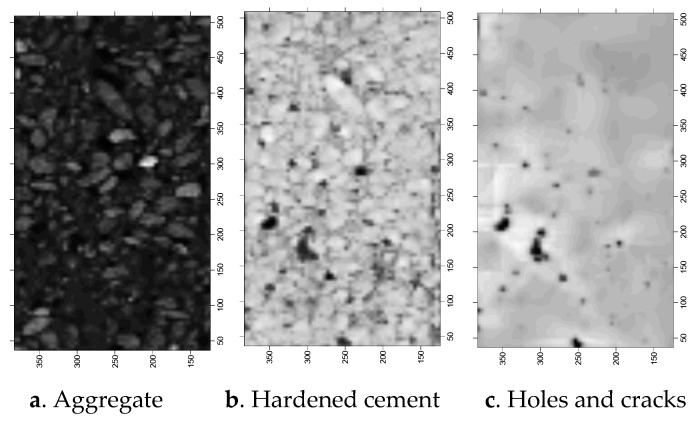
The images of each partition of CT image.

**Figure 10 materials-12-04070-f010:**
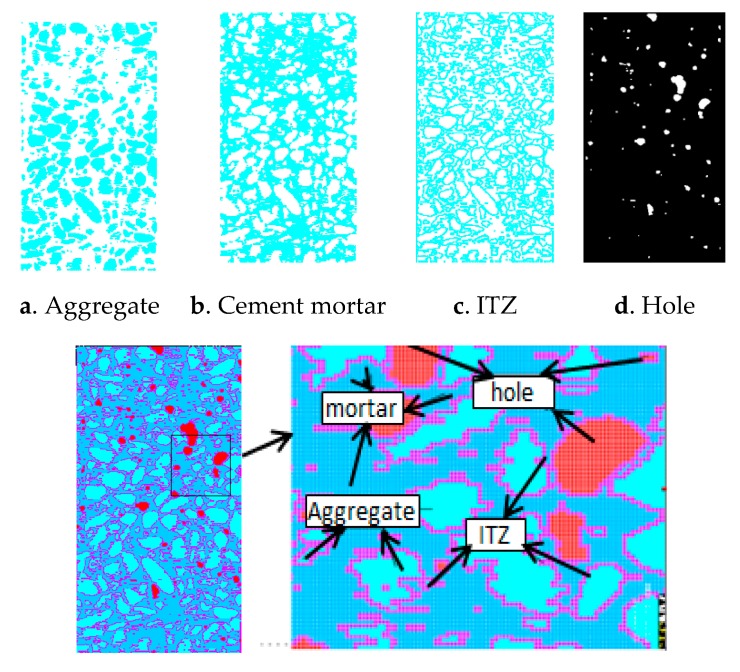
Structure random numerical concrete model.

**Figure 11 materials-12-04070-f011:**
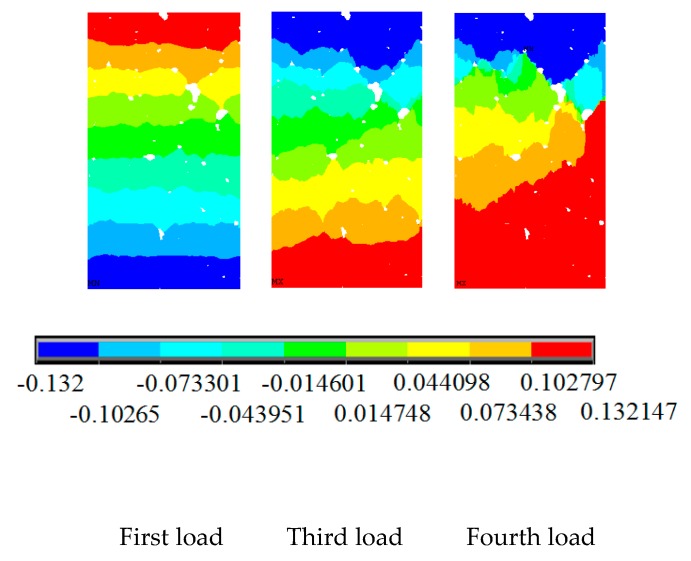
Axial displacement nephogram of concrete under uniaxial compression.

**Figure 12 materials-12-04070-f012:**
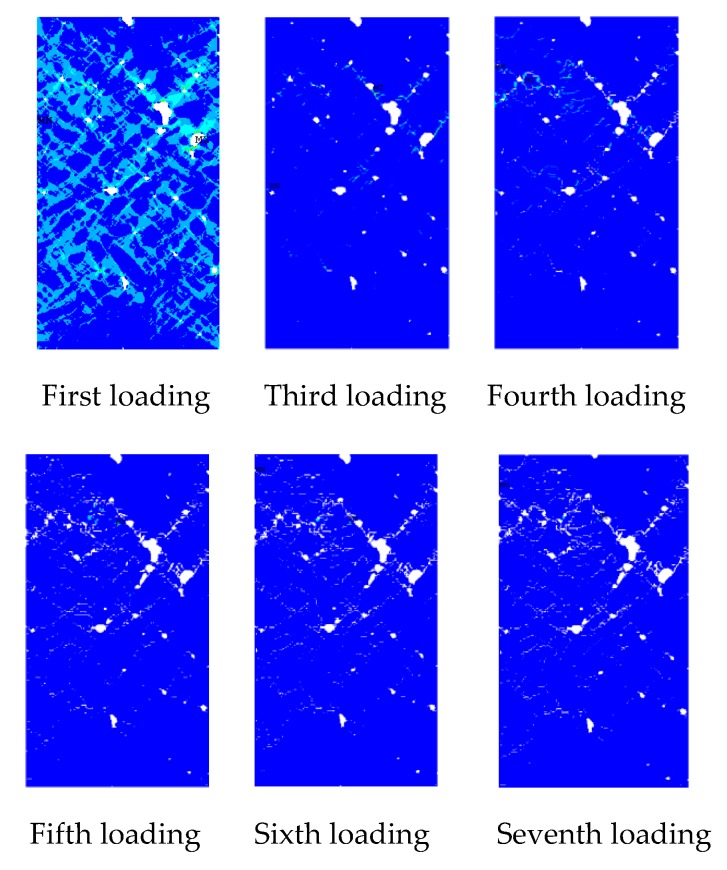
Nephogram of the first principal stress under uniaxial compression load.

**Figure 13 materials-12-04070-f013:**
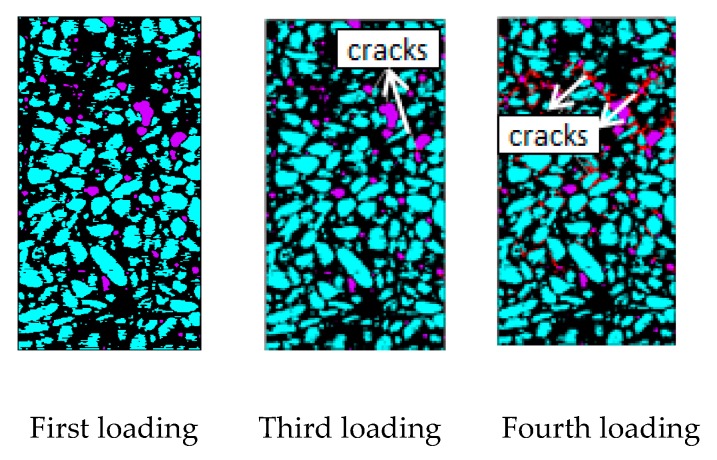
Crack growth diagram under uniaxial compression load.

**Figure 14 materials-12-04070-f014:**
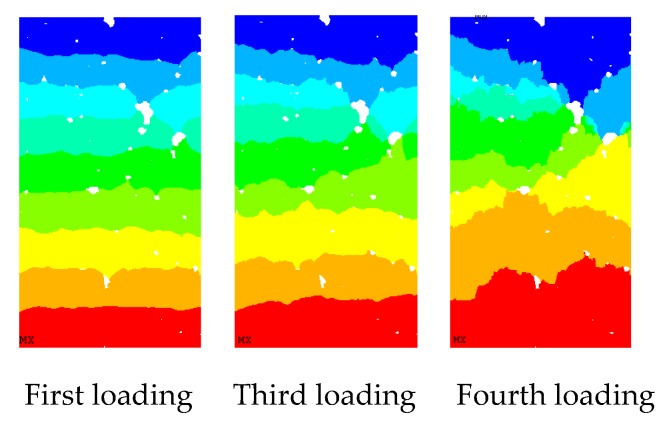
Axial displacement nephogram of concrete under uniaxial tensile load.

**Figure 15 materials-12-04070-f015:**
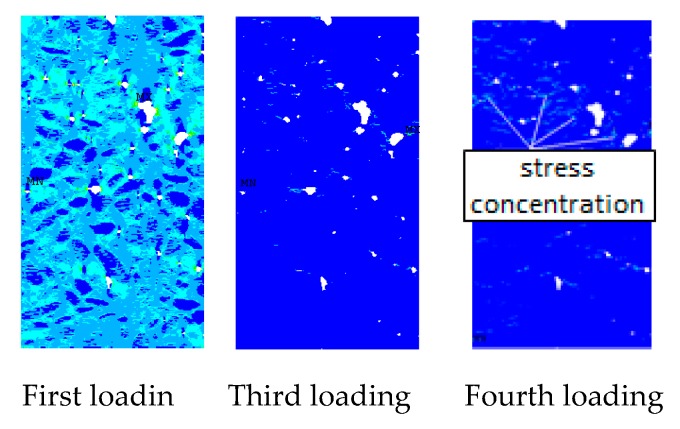
First main strain contours of concrete under uniaxial tensile load.

**Figure 16 materials-12-04070-f016:**
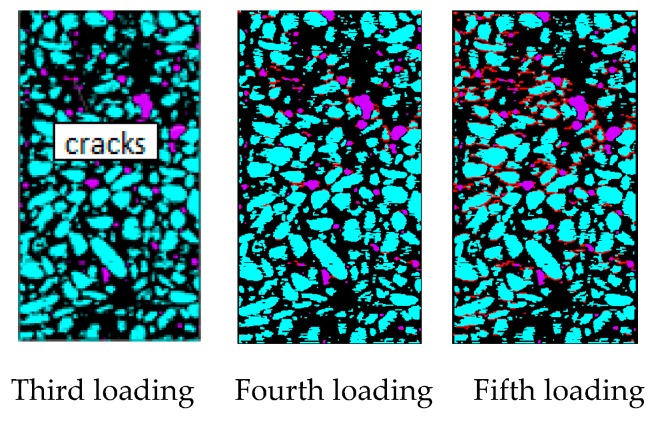
Crack growth under uniaxial tensile load.

**Table 1 materials-12-04070-t001:** Cumulative counts of resolution cell in different thresholds.

Threshold	Cumulative Probability	Threshold	Cumulative Probability	Threshold	Cumulative Probability
0.668	1.375	0.718	2.593	0.833	22.237
0.673	1.566	0.723	2.926	0.838	22.237
0.678	1.566	0.728	2.926	0.843	30.538
0.683	1.767	0.733	3.326	0.848	30.538
0.688	1.999	0.738	3.326	0.853	41.837
0.693	1.999	0.808	9.819	0.858	41.837
0.698	2.306	0.813	12.267	0.863	54.751
0.703	2.306	0.818	15.937	0.868	54.751
0.708	2.593	0.823	15.937	0.873	66.271
0.713	2.593	0.828	15.937	0.878	66.271

**Table 2 materials-12-04070-t002:** Material parameters of concrete components.

Material	Elastic Modulus (GPa)	Poisson’s Ratio	Tensile Strength (MPa)	Density (kg/m^3^)
Aggregate	58.731	0.2407	9.25	2800
Motor	17.458	0.1960	2.78	2200
Interface	13.967	0.2000	1.56	2000

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
