# Peer review of "Numerical Reconstruction Model and Simulation Study of Concrete Based on Damaged Partition Theory and CT Number"

_materials, 2019, doi:10.3390/ma12244070_

Round 1

Reviewer 1 Report

Present study focuses on the computerized tomography(CT) test of concrete under uniaxial compression conditions. In the analysis, a probabilistic statistical method for determining threshold values is proposed, and a CT test diagram is obtained and divided into aggregate, hardened cement and hole-crack areas. Current structure of the paper and the presented content is sufficient for the publication in this journal, and worthwhile for the readers. Minor spell checks are required in few places and can bit more focus on the conclusion section to stick with the new findings rather reporting experimental observations only.

Author Response

  Dear reviewer

we are very grateful for your constructive comments and suggestions. According with your suggestions, we amended the relevant part in manuscript. The following is a point-to-point response to your comments.

 Firstly,Spelling of whole text has been checked and modified. Secondly, The conclusion was changed also by adding new findings(Page 17,lines 513-522)

With kindest regards,

Yours sincerely,

Jianyin Fang, You Pan, Faning Dang , Xiyuan Zhang , Jie Ren ,Na Li

Reviewer 2 Report

The paper deals with concrete geometry reconstruction at meso-scale from CT scan of damaged specimen. The approach partition the image into aggregates, hardened cement paste and holes. Based on this geometry, damage mechanics simulations are performed and damage evolution is analyzed.

Strengths of this work include tackling the whole analysis, from scan of real specimen to numerical simulations.

However, several major weaknesses are present. First one is regarding the global quality of English which is poor, starting from the title. Introduction deals only with the scanning procedure, but mentions nothing regarding previous researches conducted on damage evolution and numerical analysis of meso-scale concrete structures. Furthermore, there is a lack of references in these sections, including some of internationally recognized authors. Figures are numerous, but legends are often unclear as well as their description.

On the technical side, some major improvements are required.

In section on damage evolution theory, do you really deal with damage evolution or (to my point of view) to identification of different areas with various damage level. Furthermore, is this method published for the first time or re-used for damaged concrete applications ?

In section 4, everything is based on the results of table 1. However, results are rather strange, e.g. big jump in cumulative probability between 0.813 and 0.818 thresholds, while none between 0.818-0.823-0.828. This is repeated several times later on. I guess you are looking beyond the precision of the CT-scan and/or grid resolution.

Section 5 lacks of details (e.g. ENVI, DICOM, which Fortran program, kill unit function (kill element approach ?)) and some comments are rather obvious (material parameters determine the accuracy of the numerical results...). The authors perform numerical simulation in both traction and compression, but in material parameters, same ones are use for both loadings. How do you distinguish that concrete behaves badly in tension ?

The most critical drawback is regarding the damage model. Where does this come from (reference ?), how are material parameters identified (I guess from numerical simulation results => no predictive effect !). Furthermore, the use of a damage model requires a regularization method. This is never mentioned. Therefore, mesh size dependence is expected but never discussed. Illustrations refer to different loadings, but these are never quantitatively described (applied displacement ? value ? ). Comparison of cracks pattern obtained numerically with experimental ones is maybe the most interesting part, but far too simple.

Author Response

Dear reviewer

We are very grateful for your constructive comments and suggestions. According with your suggestions, we amended the relevant part in manuscript.

With kindest regards,

Yours sincerely,

Jianyin Fang, You Pan, Faning Dang , Xiyuan Zhang , Jie Ren ,Na Li

Reviewer 3 Report

This paper uses the damaged partition method to reconstruct the structure random numerical concrete model, with the pixel points of CT scan as the unit. The numerical simulation test is compared with the physical CT test. 

The paper is well written and the main ideas well developed: in my opinion the article should be accepted as it is.

Author Response

Dear reviewer

We are very grateful for your constructive comments and suggestions.

Thank you very much

With kindest regards,

Yours sincerely,

Jianyin Fang, You Pan, Faning Dang , Xiyuan Zhang , Jie Ren ,Na Li

Reviewer 4 Report

The manuscript is interesting and well presented. Some minor comments:

English language needs improvements in some phrases Strengthen your conclusions by describing the potential applications of your research to the construction industry Mistake at your references 1. 1 etc

Author Response

Dear reviewer

We are very grateful for your constructive comments and suggestions. According with your suggestions, we amended the relevant part in manuscript. The following is a point-to-point response to your comments.

The author revised the spelling of the whole text, and the conclusion and references are also revised .(Page17, line 523-529).

With kindest regards,

Yours sincerely,

Jianyin Fang, You Pan, Faning Dang , Xiyuan Zhang , Jie Ren ,Na Li

Round 2

Reviewer 2 Report

Introduction has been greatly improved.

My major comment was regarding the damage model. Referenced have been added. However, there is nothing regarding the major concerns, i.e. use of a regularization method, mesh dependency,... I therefore expect some improvements on this point.

Author Response

Dear reviewer,

We are very grateful for your constructive comments and suggestions. We have amended the manuscript according to your suggestions. ( Page 17 , lines 16-19) The following is a point-to-point response to your comments.

(1)My major comment was regarding the damage model. Referenced have been added. However, there is nothing regarding the major concerns, i.e. use of a regularization method, mesh dependency,... I therefore expect some improvements on this point.

Answer: Your suggestion is greatly appreciated.

The calculation results of the traditional model have significant mesh size dependence in the numerical simulation study,and the regularization method is often used to correct it. This study focuses on the reconstruction of concrete model which based on CT number and damage evolution theory.

The size of the element is determined by the size of the resolution unit in the CT scan, so the dependency of the mesh size is not considered in the process of modeling.

Many researchers [1~5] have studied the damage evolution of concrete by using the double-fold damage constitutive model. Although the mesh dependence is not considered in these studies, the numerical experimental results are in good agreement with the physical experimental results. Therefore, the model is used to simulate the crack evolution process of concrete based on the reconstructed numerical concrete model. It is found that the crack evolution law of concrete obtained by the reconstructed numerical concrete model is similar to the results of physical CT test, and it indicates that the reconstructed model in this paper has certain research and application value.

However, the mesh dependence problem is not considered in the numerical simulation, which is a defect of this study, author is going to focus on the analysis and solution of this problem in following research. Hope you can continue to pay attention to my following research.

References:

[1] MA Huaifa, CHEN Houqun, Yang Changlu. Numerical tests of meso-scale damage mechanism for full graded concrete under complicated dynamic loads. CHINA CIVIL ENGINEERING JOURNAL. 2012, 7, 175-182.

[2] HE Jian-tao, MA Huaifa, CHEN Houqun. Research review on concrete damage constitutive theory. Advances in Science and Technology of Water Resources. 2010, 3, 89-94.

[3] Tian Ruijun, Du Xiuli, Peng Yijiang. Numerical Simulation on Compression Failure Process of Concrete and Size Effect. Industrial Construction. 2008, 38, 68-73.

[4] Tian Ruijun, Du Xiuli, Peng Yijiang. Numerical Simulation on Compression Failure Process of Concrete and Size Effect. Industrial Construction. 2008, 38, 68-73.

[5] CHEN Hou-qun , MA Huai-fa , LI Yun-cheng. Influence of random aggregate shapes on flexural strength of dam concrete. Journal of China Institute of Water Resources and Hydropower Research. 2007, 5, 241-246.